# Application of Thermally Fluorinated Multi-Wall Carbon Nanotubes as an Additive to an Li_4_Ti_5_O_12_ Lithium Ion Battery

**DOI:** 10.3390/nano13060995

**Published:** 2023-03-09

**Authors:** Seongmin Ha, Seo Gyeong Jeong, Chaehun Lim, Chung Gi Min, Young-Seak Lee

**Affiliations:** 1Department of Chemical Engineering and Applied Chemistry, Chungnam National University, Daejeon 34134, Republic of Korea; haseongmin93@cnu.ac.kr (S.H.);; 2Institute of Carbon Fusion Technology (InCFT), Chungnam National University, Daejeon 34134, Republic of Korea

**Keywords:** multi-walled carbon nanotubes (MWCNTs), fluorinated carbon (CF_x_), thermal fluorination, Li_4_Ti_5_O_12_ (LTO), lithium ion batteries (LIBs)

## Abstract

In this study, multi-walled carbon nanotubes (MWCNTs) were modified by thermal fluorination to improve dispersibility between MWCNTs and Li_4_Ti_5_O_12_ (LTO) and were used as additives to compensate for the disadvantages of LTO anode materials with low electronic conductivity. The degree of fluorination of the MWCNTs was controlled by modifying the reaction time at constant fluorination temperature; the clear structure and surface functional group changes in the MWCNTs due to the degree of fluorination were determined. In addition, the homogeneous dispersion in the LTO was improved due to the strong electronegativity of fluorine. The F-MWCNT conductive additive was shown to exhibit an excellent electrochemical performance as an anode for lithium ion batteries (LIBs). In particular, the optimized LTO with added fluorinated MWCNTs not only exhibited a high specific capacity of 104.8 mAh g^−1^ at 15.0 C but also maintained a capacity of ~116.8 mAh g^−1^ at a high rate of 10.0 C, showing a capacity almost 1.4 times higher than that of LTO with the addition of pristine MWCNTs and an improvement in the electrical conductivity. These results can be ascribed to the fact that the semi-ionic C–F bond of the fluorinated MWCNTs reacts with the Li metal during the charge/discharge process to form LiF, and the fluorinated MWCNTs are converted into MWCNTs to increase the conductivity due to the bridge effect of the conductive additive, carbon black, with LTO.

## 1. Introduction

As the demand for lithium ion batteries for energy storage devices and large devices increases, the demand for improved battery performance, including the energy density, stability, and rate capacity, is increasing [1]. For anode materials, graphite is currently widely used among the carbon-based anode active materials. The graphite anode has excellent cycle characteristics and stability, because it has a reaction mechanism in which lithium ions are reversibly intercalated and deintercalated between the carbon layers. However, graphite forms a solid electrolyte interface (SEI) layer due to a low lithium intercalation potential that is close to the lithium deposition potential. The formation of such an SEI layer causes the irreversible consumption of Li^+^ and deposition of lithium metal. In addition, due to the volume expansion of graphite (9–13%) during the charge/discharge process, the electrical contact between the particles gradually weakens, resulting in a decrease in the cycling performance [2,3,4,5,6]. Accordingly, many studies have been conducted using an alternative anode materials with high safety and excellent cycling stability, and Li_4_Ti_5_O_12_ (LTO) is attracting attention as an alternative anode material. LTO has a spinel structure, which has better structural stability compared to graphite, and it shows a plateau operating voltage at approximately 1.5 V compared to lithium. This voltage is lower than the reduction potential of most electrolytes and does not form an SEI layer [7,8]. However, despite these advantages, LTO has a problem regarding its discharge rate characteristics due to its low electrical conductivity. To compensate for these shortcomings, recently, many studies have been conducted to improve the electrical conductivity of LTO. Methods for improving the electrical conductivity of LTO by manufacturing or coating a composite with a conductive material, such as a carbon-based nanomaterial, and the synthesis of nanosized LTO have been studied [9,10,11,12,13,14,15]. In addition, the conductivity and cycle stability were improved by doping ions, such as Na^+^, K^+^, Mg2^+^, Ni^2+^, Cu^2+^, Al^3+^, V^5+^, F^−^, and Br^−^, into the LTO anode [16,17,18,19,20,21,22].

The application of MWCNTs in the form of bundles of small particles, mainly for use as conductive nanocarbons, is difficult due to its dispersion problem. Since MWCNTs that are not smoothly dispersed and aggregated are the main cause of the decrease in electrical conductivity, the electrochemical characteristics may be degraded. To resolve the dispersion problem, we determined that the doping of a fluorine functional group on the surface of a carbon material was effective [23,24,25,26].

In this study, to investigate the electrochemical performance of MWCNTs depending on the roles of fluorine functional groups and their degrees of fluorination to expand the application of MWCNTs, the MWCNTs were mixed with carbon black and used as a conductive material for LIBs based on LTO anode materials. In addition, the mechanism of the structure of fluorinated MWCNTs in relation to the electrochemical performance of lithium-ion batteries was studied.

## 2. Experimental

### 2.1. Preparation of Fluorinated Multi-Walled Carbon Nanotubes

The surfaces of multi-walled carbon nanotubes (MWCNTs, inner diameter and length of 6–13 nm and 2.5–20 μm, respectively, Sigma-Aldrich, Seoul, Korea) were treated using a fluorination apparatus consisting of a nickel reactor, a vacuum pump, a nickel boat and a buffer tank connected to gas cylinders. The sample was loaded into the nickel reactor using the nickel boat and vacuum pumped at 100 °C for 2 h to remove impurities, such as water. The thermal fluorination treatment was carried out at 300 °C for 6 and 12 h at 1 bar using fluorine gas (99.8% purity, Messer Grieheim GmbH, Bad Soden, Germany). After fluorination, the samples were degassed to remove the unreacted gas. The pristine samples were named as MWCNTs, and fluorinated samples were named as F-MWCNT_6 and F-MWCNT_12, according to the fluorination reaction time.

### 2.2. Characterization

The crystal structure was characterized by powder X-ray diffraction (XRD, Brucker D8 Advance diffractometer with Cu Kα radiation). Chemical components were characterized by X-ray photoelectron spectroscopy (XPS, K-Alpha XPS instrument, ThermoFisher Scientific, East Grinter, UK). The microstructures of the samples were examined by scanning electron microscopy (SEM, Hitachi S-4800, Tokyo, Japan) and transmission electron microscopy (TEM, Tecnai G2 F30, Hillsboro, OR, USA).

### 2.3. Preparation of Electrode and Coin Cell

Fluorinated MWCNTs were used, instead of a small amount of carbon black, when fabricating the electrodes. The electrode was produced by mixing Li_4_Ti_5_O_12_ (LTO, D50: 0.8-1.9 µm, Toshiba, Tokyo, Japan), the MWCNTs (or F-MWCNTs), carbon black (Super P) and polyvinylidene fluoride (PVDF) and dissolving them in N-methyl-2-pyrrolidone (NMP) with a weight ratio of 8:(0.25:0.75):1, respectively. Then, the slurry was spread onto an Al foil and dried at 80 °C for 12 h to remove the solvent in a vacuum environment. The electrolyte used was 1 M LiPF6 dissolved in an ethylene carbonate (EC) and diethyl carbonate (DEC) mixture (1:1 vol.%). A 2032-type lithium coin half-cell consisting of the fabricated anode, lithium metal, electrolyte and separator (Celgard 2400, Celgard Co. ltd., Charlotte, NC, USA) was assembled in an argon-filled glove box with water and oxygen contents of less than 0.5 ppm. After sitting for 24 h, the C-rate performance and cycling performance of the cell was evaluated in a voltage range of 1.0 to 2.5 V using a multichannel battery cycler (PESC05-0.1, Won-A Tech, Seoul, Korea). The C-rate characteristic was measured from 0.2 C to 15.0 C, and the cycle performance was assessed at current densities of 5.0 and 10.0 C. In addition, electrochemical impedance spectroscopy (EIS) was performed after the 150th cycle to determine the resistance of the charge transfer reaction occurring at the electrode/electrolyte interface and the diffusion of lithium ions in the electrode.

## 3. Results and Discussion

### 3.1. Effect of Thermal Fluorination on the Structure of MWCNTs

The chemical compositions of the MWCNT and F-MWCNT samples are shown in Figure 1, and the surface element contents are shown in Table 1. As shown in Figure 1a, C, O and F appeared at approximately 284.5, 533 and 687 eV [27]. As fluorination proceeded, the peak intensity of F 1s became stronger, and the peak intensity of C 1s became weaker. As shown in Figure 1 b–d and Table 2, the C 1s spectra of the MWCNTs were divided into four peaks: the C=C bond (284.4 eV), C–C bond (285.2 eV), C–O bond (286.5 eV) and C=O bond (286.6 eV) [28]. The oxygen functional groups (C-O, C=O bond) decreased during fluorination due to the initial presence of the oxygen functional groups [29]. Additionally, the F-MWCNT samples showed semi-ionic C–F (288.8 eV), C–F2 (290.0 eV) and C–F3 bonds (294.5 eV) [30]. The semi-ionic C–F bond in F-MWCNTs_6 was much larger than that in F-MWCNT_12. The C–F_2_ and C–F_3_ bonds were mainly present in F-MWCNTs_12. During the discharge process, the F-MWCNTs_6 sample had more activated C–F electrons than the F-MWCNTs_12 sample; this affected the charge/discharge capacity. The F 1s spectra of the F-MWCNT samples were confirmed by the semi-ionic bond at 688.34 eV and the covalent C–F bond at 691.6 eV [31]. A semi-ionic bond maintains a good conductivity of carbon, while a covalent C–F bond greatly reduces the conductivity, leading to insulating properties [32]. As shown in Equation (1), the amount of carbon converted in the electrochemical reaction between the F-MWCNTs and lithium was determined to be greater in the case of F-MWCNTs_6, which had more semi-ionic C–F bonds than F-MWCNTs_12.

The XRD patterns of the MWCNT and fluorinated samples are shown in Appendix A. The peak at 2θ = 24.66° can be allocated to the reflection of (002) of the graphite structure, and a new phase is present after fluorination due to the peaks at the center of approximately 13° and 42°, which are known to be influenced by the diffraction of the lattice planes (001) of fluorinated carbon materials. The (002) peak for F-MWCNT gradually decreases as the fluorination reaction time (i.e., degree of fluorination) increases and finally disappears after 12 h of the reaction time, while the (001) and (100) peaks assigned to the fluorination phase monotonically increase. These trends show that fluorination occurs from the outside to the inside, and an increasing number of carbon layers are fluorinated as the fluorination time increases. Figure 2 shows the TEM images of the MWCNT and F-MWCNT samples. No significant change occurred in F-MWCNTs_6, even after thermal fluorination. Its structure was similar to that of the MWCNTs. In contrast, F-MWCNTs_12 showed an amorphous and disordered layer structure, in addition to the graphite layer caused by the per-fluorination of the MWCNTs, and this was also indicated by the partial bursting (Appendix A) [33,34]. As mentioned above, since F-MWCNTs_12 was in the form of agglomerated carbon due to bursting, it remained as agglomerated carbon rather than an MWCNT structure during the electrochemical reaction; therefore, the F-MWCNTs_6 sample that maintained its MWCNTs structure had a better performance. In Appendix A, the dispersion between LTO and the MWCNTs (and F-MWCNT samples) was confirmed through SEM analysis. The MWCNTs were partially found on the LTO surface, and the F-MWCNT samples were found to be completely distributed on the LTO surface. This result showed that the dispersibility was improved due to the interaction of LTO with Li^+^, caused by the strong electronegativity of the fluorine-doped MWCNTs. Thus, a more uniform dispersion was achieved.

### 3.2. Electrochemical Performance of LTO with Added F-MWCNTs

Figure 3a shows the activation curves of LTO with the added MWCNTs, F-MWCNTs_6 and F-MWCNTs_12. During the first discharge process, double potential plateaus were observed due to different reactions of the LTO with the added MWCNTs and fluorinated MWCNTs. The plateau levels at approximately 2.30 V or higher could be explained based on the irreversible reaction mechanism [35,36] between fluorinated carbon (CF_x_) and lithium that occurs during the lithium primary battery reaction, as shown in Equation (1) below.
xLi+ + CF_x_ + xe^−^ → x LiF + C(1)

Therefore, the fluorinated MWCNTs were converted into MWCNTs through the abovementioned reaction and bridged between the LTO and conductive carbon (carbon black) particles. Figure 3b,c and Appendix A show the rate performance and galvanostatic charge/discharge curves of the LTO with the added MWCNTs and F-MWCNT samples at various current rates of 0.2 to 15.0 C. The capacities at various rates of LTO with the added MWCNTs were 150.0, 145.5, 144.3, 121.2, 62.3 and 1.1 mAh g^−1^; those of F-MWCNTs_6 were 161.2, 158.4, 158.3, 149.8, 125.1 and 104.8 mAh g^−1^; and those of LTO with the added F-MWCNTs_12 were 161.3, 155.2, 148.9, 138.5, 103.2 and 46.8 mAh g^−1^.

Compared to the LTO with the added MWCNTs, the F-MWCNT samples showed a high performance at all the operating rates, and in particular, significant performance differences were confirmed at the rates of 10.0 and 15.0 C. In addition, a capacity difference of approximately 2.2 times was determined for LTO with added F-MWCNTs_6 and F-MWCNTs_12 at 15.0 C. These results indicated that the bursting due to the per-fluorination of the MWCNTs and the carbon formed during the first discharge could not act as a bridge between the LTO and conductive carbon due to aggregation. Figure 4a,b shows the long-term cycling performances of LTO with the added MWCNTs, F-MWCNTs_6 and F-MWCNTs_12 at 5.0 C and 10.0 C. After 150 cycles at 5.0 C, the LTO with added F-MWCNTs_6 retained a high capacity of 129.9 mAh g^−1^. Even after 250 cycles at 10.0 C, the LTO with added F-MWCNTs_6 showed a capacity of 116.8 mAh g^−1^, which was higher than those of the LTO with added MWCNTs and F-MWCNTs_12. Furthermore, the LTO with added F-MWCNT samples showed smaller polarizations than the LTO with added MWCNTs, as shown in Figure 4c,d.

As a result, the LTO with added F-MWCNTs played a role in improving the conductivity and greatly reducing the resistance of the electrode. In addition, Figure 5 shows the EIS analysis results for the LTO with the added MWCNT and F-MWCNT samples. In the Nyquist plot, the two compressed semicircles in the high- and middle-frequency regions correspond to the impedance of the solid interphase layer and the charge transfer, respectively, and the slant in the low-frequency region is related to the Warburg impedance of lithium ion diffusion. The equivalent circuit consists of the ohmic resistance R_S_, charge transfer resistance R_CT_, SEI layer resistance R_SEI_, two constant phase elements CPE_1_ and CPE_2_ that model the defect capacitor behaviors of the SEI layer and double layer, respectively, and a Warburg diffusion element Z_W_, as shown in Figure 5. The resistance of the LTO with the added F-MWCNT samples was lower than that of the LTO with added MWCNTs, and the conductivities were improved due to the small semicircle. The decrease in R_CT_ was due to self-stabilization during repeated charge/discharge processing [37,38]. The small R_CT_ values of these LTO with added F-MWCNT samples showed an excellent rate and cycling performance. As described above, the LTO with added F-MWCNTs_12 had more covalent C–F bonds than the LTO with added F-MWCNTs_6; therefore, the LTO with added F-MWCNTs_12 did not serve as a bridge, because less carbon was converted during the CF_x_ discharge process. In addition, this difference potentially occurred because there were many carbons that burst or aggregated due to the per-fluorination of F-MWCNTs_12. However, the F-MWCNT samples showed a better electrochemical performance than the MWCNTs.

## 4. Conclusions

MWCNTs were modified by thermal fluorination to improve the dispersibility between MWCNTs and LTO and were used as additives to compensate for the disadvantages of LTO anode materials with low electronic conductivity. It can be confirmed that as the fluorination proceeds, fluorine-doped MWCNT reacts with the Li^+^ of LTO, resulting in a more homogeneous dispersibility due to the strong electronegativity of fluorine. It was verified the optimized fluorinated MWCNTs did not significantly change in terms of structure during thermal fluorination, and many semi-ionic C–F bonds were formed on the MWCNTs. In addition, the electrochemical performances of the materials confirmed their capacity of 104.8 mAh g^−1^, even at a high rate of 15.0 C, and they exhibited a low resistance, indicating that the electronic conductivity was improved. These findings can be ascribed to the fact that the semi-ionic C–F bond of the fluorinated MWCNTs reacts with the Li metal during the charge/discharge process to form LiF, and the fluorinated MWCNTs are converted into MWCNTs so as to increase their conductivity due to the bridge effect of the conductive additive, carbon black, with LTO.

## Figures and Tables

**Figure 1 nanomaterials-13-00995-f001:**
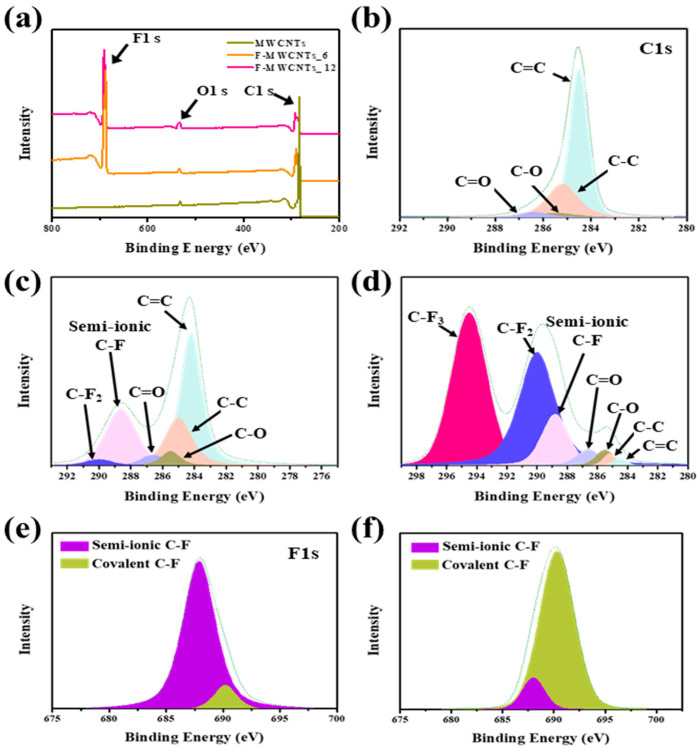
(**a**) X-ray photoemission spectroscopy (XPS) profiles of MWCNT, F-MWCNT 6 and F-MWCNTs_12. (**b**–**d**) C 1s spectra of the MWCNTs, F-MWCNTs_6 and F-MWCNTs_12. (**e**,**f**) F 1s spectra of the MWCNTs, F-MWCNTs_6 and F-MWCNTs_12.

**Figure 2 nanomaterials-13-00995-f002:**
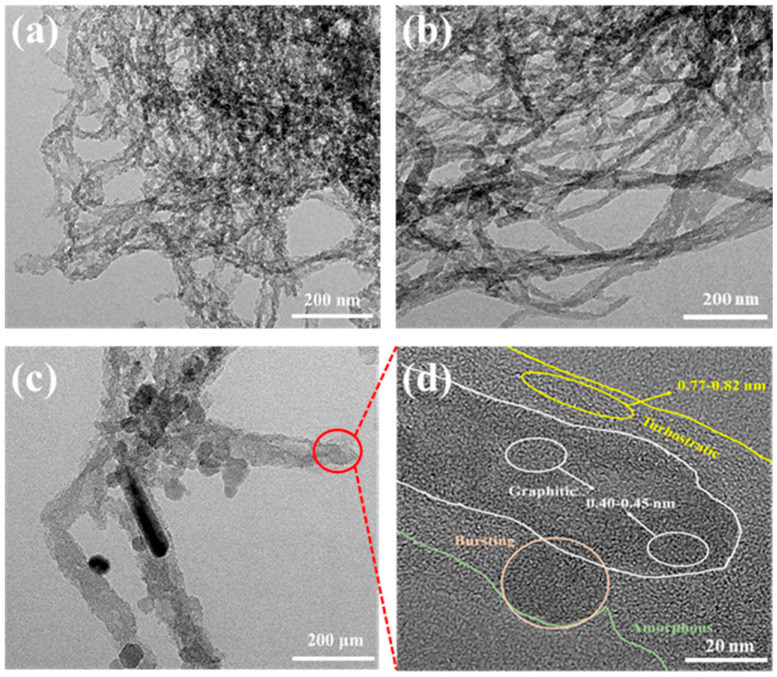
TEM images of (**a**) MWCNTs, (**b**) F-MWCNTs_6 and (**c**) F-MWCNTs_12 and (**d**) microstructure of F-MWCNTs_12.

**Figure 3 nanomaterials-13-00995-f003:**
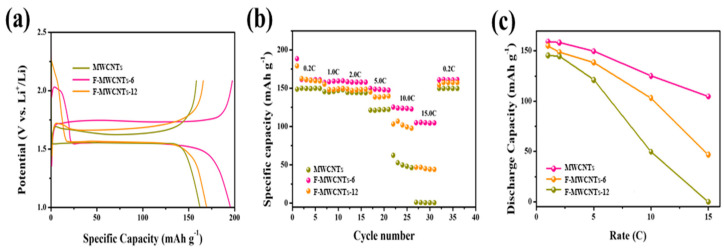
(**a**) The charge/discharge curves of the first discharge of LTO with the added MWCNTs MWCNTs, F-MWCNTs_6 and F-MWCNTs_12. (**b**,**c**) Rate performance of LTO with added MWCNTs, F-MWCNTs_6 and F-MWCNTs_12.

**Figure 4 nanomaterials-13-00995-f004:**
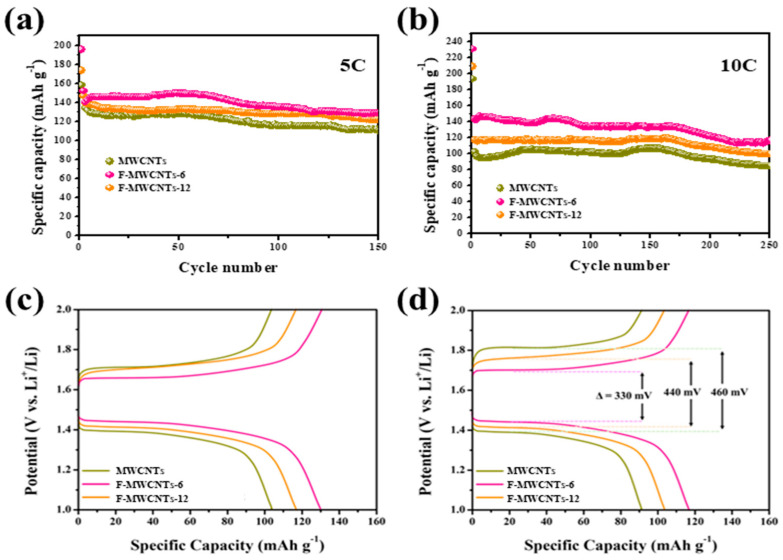
Cycling performance of LTO with added MWCNTs, F-MWCNTs_6 and F-MWCNTs_12 at (**a**) 5 and (**b**) 10 C. The corresponding (**c**) 100th and (**d**) 200th galvanostatic charge/discharge curves of LTO with added MWCNTs, F-MWCNTs_6 and F-MWCNTs_12.

**Figure 5 nanomaterials-13-00995-f005:**
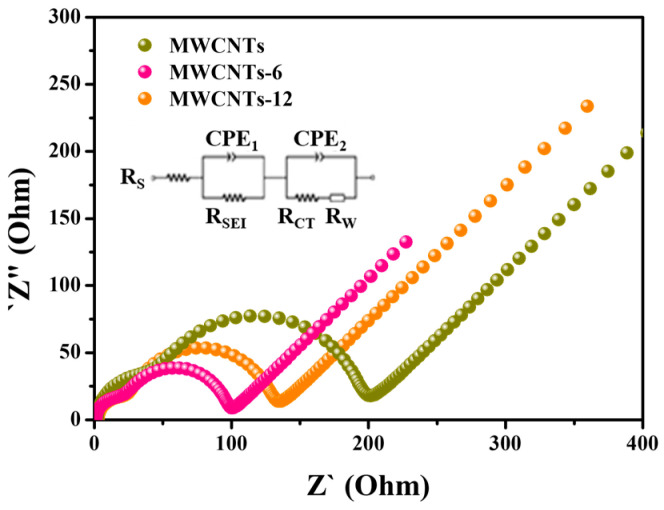
AC impedance spectra after 200 cycles of LTO with added MWCNTs, F-MWCNTs_6 and F-MWCNTs_12.

**Table 1 nanomaterials-13-00995-t001:** Surface elemental compositions of MWCNTs, F-MWCNTs_6 and F-MWCNTs_12 based on the XPS spectra.

	Element Content (at.%)		
	C	F	O
MWCNTs	97.42	-	2.58
F-MWCNTs_6	59.32	38.13	2.55
F-MWCNTs_12	53.64	43.83	2.54

**Table 2 nanomaterials-13-00995-t002:** Peak parameters for the C 1s components of MWCNTs, F-MWCNTs_6 and F-MWCNTs_12.

Assignment	Binding Energy(eV)	Concentration (%) of Each Sample		
		MWCNTs	F-MWCNTs_6	F-MWCNTs_12
C=C	284.4	65.03	44.75	0.75
C-C	285.2	26.88	20.04	1.19
C-O	285.5	4.40	4.19	2.3
C=O	286.6	3.69	3.30	3.5
Semi-ionic C–F	288.8	-	25.20	11.84
C–F_2_	290	-	2.52	37.84
C–F_3_	294.5	-	-	42.58

## Data Availability

Data are contained within the article.

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
