# Peer review of "Application of Thermally Fluorinated Multi-Wall Carbon Nanotubes as an Additive to an Li4Ti5O12 Lithium Ion Battery"

_nanomaterials, 2023, doi:10.3390/nano13060995_

Round 1
Reviewer 1 Report
The article written by S. Ha and co-authors in my first view dealing with the utilization of MWCNT, their modification with fluorine ions and connection as a conductive material with LTO anode material. This was my first vision, after reading the title, abstract, and introduction. After carefully reading the whole manuscript, I feel completely lost, and see that experimental and discussion parts are mainly dedicated to MWCNT and F-MWCNT materials. Something is missing. This title of the manuscript “Effect of thermal fluorination of carbon nanotubes as additives on the electrochemical performance of a Li4Ti5O12 lithium-ion battery” is not expressing this what is inside.
Please pay attention of that article before re-submitting it to the Nanomaterials journal. The idea is interesting, but where are the results? XRD, SEM, XPS, electrochemical measurements and other studies should be presented in the main text for the LTO material modified with MWCNT, F-MWCNT and discussed in details. That works should be helpful: Diam. Relat. Mater. 113 (2021) 108276; Solid State Ionics 384 (2022) 116005.
Supplementary Materials are not uploaded, so the reviewer should trust in this what is not visible?
At this moment I must to reject the article, but would like to believe that the authors read and check it carefully before submission to the journal as well as address my suggestions.
Author Response
Point 1: After carefully reading the whole manuscript, I feel completely lost, and see that experimental and discussion parts are mainly dedicated to MWCNT and F-MWCNT materials. Something is missing. This title of the manuscript “Effect of thermal fluorination of carbon nanotubes as additives on the electrochemical performance of a Li4Ti5O12 lithium-ion battery” is not expressing this what is inside.
Response 1: Thank you for the careful examination. This paper is published in "Fluorinated Nanocarbons and Their Applications II," a special issue of nanomaterials. So we covered the fluorination of nanocabon and its application to suit the subject of the special issue. Therefore, in this paper, the physical properties of F-MWCNTs manufactured mainly according to the fluorination time were investigated and analyzed, and these samples were applied to Li4Ti5O12 lithium-ion batteries, focusing on the effect of these fluorinated samples on performance. According to the reviewer's opinion, the title has been modified as follows to suit the content of the text.
“Application of thermally fluorinated multiwall carbon nanotube as an additive to Li4Ti5O12 lithium-ion battery”
Point 2: The idea is interesting, but where are the results? XRD, SEM, XPS, electrochemical measurements and other studies should be presented in the main text for the LTO material modified with MWCNT, F-MWCNT and discussed in details. That works should be helpful: Diam. Relat. Mater. 113 (2021) 108276; Solid State Ionics 384 (2022) 116005.
Response 2: As I answered the first question, We considered the applicability of thermally fluorinated multi-walled carbon nanotubes as an additive to the Li4Ti5O12 lithium-ion battery. Therefore, although there was not much analysis on the mixture of LTO and additives such as MWCNT and F-MWCNT, SEM data for these are shown in Figure S3, and electrochemical data are shown in Figure 3-5 and Figure S4.
Point 3: Supplementary Materials are not uploaded, so the reviewer should trust in this what is not visible?
Response 3:. We uploaded the supplementary data when we first submitted our manuscript, and we checked it on the posting site. I don't know if there's a technical problem, but please check it again.
Once again, I would like to thank you for your careful review.

Reviewer 2 Report
In this study, the authors use MWCNT to improve dispersibility be-tween MWCNTs and LTO. I hope that the following comments are useful in improving the paper, and I dearly hope that the authors will take the time to revise and take these on board.
1. Some definitions of abbreviations are missing, please complete them.
2. In XPS profiles, the corresponding chemical bond should be added in the Figure 1 rather than "C1,C2,C3,C4..."
3. The performace of electrode without the super P should be exhibited.
4. Please check the MS. There are some grammatical errors that need to be corrected.
Author Response
In this study, the authors use MWCNT to improve dispersibility be-tween MWCNTs and LTO. I hope that the following comments are useful in improving the paper, and I dearly hope that the authors will take the time to revise and take these on board.
Point 1: Some definitions of abbreviations are missing, please complete them.
Response 1: Thank you for your comment. I explained all the abbreviations.
Point 2: In XPS profiles, the corresponding chemical bond should be added in the Figure 1 rather than "C1,C2,C3,C4..."
Response 2: Thank you for your thoughtful comment. As the reviewer mentioned, the chemical bond of C1s peak was changed as shown in Figure 1 and Table 2.
Point 3: The performace of electrode without the super P should be exhibited.
Response 3: Thank you for your careful consideration. As you already know, when an electrode is manufactured using 100% MWCNT without Super P as a conductive additive, the battery performance is not good due to the low conductivity of the active material in the battery, so most secondary batteries are composed of active materials, binders, conductive materials, etc.
Point 4: Please check the MS. There are some grammatical errors that need to be corrected.
Response 4: Thank you for your comment. Our manuscript has been edited for proper English, grammar, punctuation, spelling, and overall style by a highly qualified native editor of American Journal Experts (certification key: 9DC6-7823-0183-FFDF-787P). Also, I modified it once again through program “Grammerly”.

Reviewer 3 Report
In this paper, the authors carefully controlled the degree of fluorination and observed the changes in structure and surface functional groups, leading to improved dispersion and electronegativity. The results showed that the Li4Ti5O12 with optimized fluorinated MWCNTs (F-MWCNTs) had a significantly higher specific capacity and improved electrical conductivity compared to with pristine MWCNTs, demonstrating F-MWCNTs potential as a conductive additive in LIBs. The mechanisms behind these improvements are clearly explained and provide a valuable contribution to the field of energy storage materials. I recommend this manuscript published in the journal of "Nanomaterials" with the following modifications.
1. Please double-check that there are several statements that use CNT instead of MWCNT.
2. In Line 139-140, “These trends show that fluorination occurs from the outside to the inside, and an increasing number of carbon layers are fluorinated as the fluorination temperature increases.” should be “...fluorination time increases”
3. In the electrochemical performance analysis section, referring to the materials as "MWCNTs or F-MWCNTs" can cause confusion and misperception among the readers, as F-MWCNTs only serve as conductive agents (0.25%) and are not the main electrode material. It would be more accurate to describe the materials as "LTO with added MWCNTs or F-MWCNTs" .
4. “Figure 3. (a) The discharge curve of the first discharge of MWCNTs, F-MWCNTs_6, and F-178 MWCNTs_12;”should be “Figure 3. (a) The first charge/discharge curves of the MWCNTs, F-MWCNTs_6, and F-178 MWCNTs_12;”
5. Please move the discussion in the caption of Figure 5 to the main text.
6. Is it possible to give the possible mechanism for the decrease of the semi-ionic C-F bond and the increase of the covalent C-F bond with the increase of the fluorination time?
Author Response
In this paper, the authors carefully controlled the degree of fluorination and observed the changes in structure and surface functional groups, leading to improved dispersion and electronegativity. The results showed that the Li4Ti5O12 with optimized fluorinated MWCNTs (F-MWCNTs) had a significantly higher specific capacity and improved electrical conductivity compared to with pristine MWCNTs, demonstrating F-MWCNTs potential as a conductive additive in LIBs. The mechanisms behind these improvements are clearly explained and provide a valuable contribution to the field of energy storage materials. I recommend this manuscript published in the journal of "Nanomaterials" with the following modifications.
Point 1: Please double-check that there are several statements that use CNT instead of MWCNT.
Response 1: Thank you for your comment. These words have been revised according to your comment.(CNT -> MWCNT)
Point 2: In Line 139-140, “These trends show that fluorination occurs from the outside to the inside, and an increasing number of carbon layers are fluorinated as the fluorination temperature increases.” should be “...fluorination time increases”
Response 2: Thank you for your careful consideration. This sentence has been modified.
Point 3: In the electrochemical performance analysis section, referring to the materials as "MWCNTs or F-MWCNTs" can cause confusion and misperception among the readers, as F-MWCNTs only serve as conductive agents (0.25%) and are not the main electrode material. It would be more accurate to describe the materials as "LTO with added MWCNTs or F-MWCNTs" .
Response 3: Thank you for your thoughtful comment. We modified all parts of the electrochemical part expressed as MWCNT or F-MWCNT to "LTO with MWCNT or F-MWCNT added".
Point 4: “Figure 3. (a) The discharge curve of the first discharge of MWCNTs, F-MWCNTs_6, and F-178 MWCNTs_12;”should be “Figure 3. (a) The first charge/discharge curves of the MWCNTs, F-MWCNTs_6, and F-178 MWCNTs_12;”
Response 4: Thank you for your careful consideration. This sentence has been modified.
Point 5: Please move the discussion in the caption of Figure 5 to the main text.
Response 5: Thank you for your comment. We revised it.
Point 6: Is it possible to give the possible mechanism for the decrease of the semi-ionic C-F bond and the increase of the covalent C-F bond with the increase of the fluorination time?
Response 6: Thank you for the good suggestion. The mechanism of C-F chemical bonding of MWCNT depending on the fluorination time at high temperature varies greatly depending on the fluorination temperature, time, and fluorine doping amount, etc, so it would be difficult in this manuscript to identify the exact mechanism in this time. I ask for your understanding.
